# Cytotoxic CD4+ T-cells specific for EBV capsid antigen BORF1 are maintained in long-term latently infected healthy donors

**Alexander C. Dowell**[1], **Tracey A. Haigh**[1], **Gordon B. Ryan**[1], **James E. Turner**[2], **Heather M. Long**[1], **Graham S. Taylor**[1]*

**1** Institute of Immunology and Immunotherapy, College of Medical and Dental Sciences, University of Birmingham, Birmingham, United Kingdom, **2** Department for Health, University of Bath, Claverton Down, Bath, United Kingdom

* G.S.Taylor@bham.ac.uk

**Data Availability Statement:** All relevant data are within the manuscript and its Supporting Information files.

## Abstract

Epstein Barr Virus (EBV) infects more than 95% of the population whereupon it establishes a latent infection of B-cells that persists for life under immune control. Primary EBV infection can cause infectious mononucleosis (IM) and long-term viral carriage is associated with several malignancies and certain autoimmune diseases. Current efforts developing EBV prophylactic vaccination have focussed on neutralising antibodies. An alternative strategy, that could enhance the efficacy of such vaccines or be used alone, is to generate T-cell responses capable of recognising and eliminating newly EBV-infected cells before the virus initiates its growth transformation program. T-cell responses against the EBV structural proteins, brought into the newly infected cell by the incoming virion, are prime candidates for such responses. Here we show the structural EBV capsid proteins BcLF1, BDLF1 and BORF1 are frequent targets of T-cell responses in EBV infected people, identify new CD8+ and CD4+ T-cell epitopes and map their HLA restricting alleles. Using T-cell clones we demonstrate that CD4+ but not CD8+ T-cell clones specific for the capsid proteins can recognise newly EBV-infected B-cells and control B-cell outgrowth via cytotoxicity. Using MHC-II tetramers we show a CD4+ T-cell response to an epitope within the BORF1 capsid protein epitope is present during acute EBV infection and in long-term viral carriage. In common with other EBV-specific CD4+ T-cell responses the BORF1-specific CD4+ T-cells in IM patients expressed perforin and granzyme-B. Unexpectedly, perforin and granzyme-B expression was sustained over time even when the donor had entered the long-term infected state. These data further our understanding of EBV structural proteins as targets of T-cell responses and how CD4+ T-cell responses to EBV change from acute disease into convalescence. They also identify new targets for prophylactic EBV vaccine development.

## Author summary

Epstein-Barr virus is a widespread herpesvirus carried by most individuals. Whilst infection is usually asymptomatic, development of a prophylactic vaccine against EBV is

**Funding:** This research was supported by the Cancer Research UK award C8781/A13174 awarded to GST. ACD, JET, and TH were salaried from this award. The funder had no role in the study design, data collection and analysis, decision to publish, or preparation of the manuscript.

**Competing interests:** The authors have declared that no competing interests exist.

desirable because of the virus's association with infectious mononucleosis in primary infection and several cancers and autoimmune diseases during long-term virus carriage. Identifying T-cell responses that can recognise newly infected B-cells at very early stages of infection may provide novel targets for T-cell vaccination. Here we characterise T-cell responses against three virus proteins, BcLF1, BDLF1 and BORF1 that, as structural proteins of the virus particle, are delivered into the cell by the infecting virus. We find that all three proteins are recognised by T-cells from infected individuals. Moreover, isolated structural antigen-specific CD4+ T-cells rapidly recognise newly infected B-cells and prevent their outgrowth *in vitro*. As reported for CD4+ T-cells against other EBV proteins, structural antigen-specific CD4+ T-cells induced by primary EBV infection have cytotoxic function. However, we also demonstrate that, unusually, this cytotoxic function is retained in memory T-cells present in long-term infected individuals. Structural antigens may therefore represent useful targets for prophylactic EBV vaccine development to induce CD4+ T-cells able to rapidly eliminate virus-infected cells.

## Introduction

Epstein Barr Virus (EBV), a gammaherpes virus that primarily infects human B-cells and epithelial cells, infects over 95% of the world's population. Commonly asymptomatic in young children, delayed infection in adolescents often results in infectious mononucleosis (IM). This usually self-limiting lymphoproliferative disease has symptoms of fever, fatigue, sore throat and lymphoadenopathy that are associated with an overactive immune response [1]. T-cells contribute to these symptoms but also play a key role in controlling the infection. IM patients have a highly expanded population of EBV-specific CD8+ T-cells accompanied by a smaller population of virus-specific CD4+ T-cells [2–8]. These T-cells target the large array of proteins expressed sequentially in the virus lytic cycle during replication, and the six nuclear antigens (EBNA1, 2, 3A, 3B, 3C and LP) and two latent membrane proteins (LMP1 and 2) that drive the life-long persistent infection EBV ultimately establishes in the host's B-cell system [8]. As primary infection resolves, viral gene expression is down-regulated and these T-cells decrease in frequency, entering long-term memory where they are essential for controlling the periodic reactivations of EBV that occur in the oropharynx to produce infectious virus to infect new hosts [8].

Most people recover from their primary EBV infection without any long-term consequences, carrying the virus for the rest of their lives. However, this belies the virus's potent growth transforming activity. EBV is associated with several malignancies estimated at 200,000 cases each year worldwide and poses a significant threat to patients with primary or secondary immunodeficiencies, such as transplant recipients receiving iatrogenic immunosuppression, that compromise T-cell control of the virus [9]. Accumulating evidence also links EBV infection to an increased risk of various autoimmune conditions, particularly multiple sclerosis [10].

Work on two other human oncoviruses, hepatitis B virus and human papillomavirus, clearly shows it is possible to prevent transmission and, in turn, reduce the incidence of virus-associated cancers using prophylactic vaccination [11,12]. A prophylactic EBV vaccine able to suppress the set-point of EBV carriage, which is associated with reduced lymphoma risk in immunosuppressed patients [13], could reduce the global burden of disease. The ultimate goal, however, should be to prevent transmission in order to achieve the most favourable risk-benefit and cost-benefit outcomes. Most EBV prophylactic vaccine research has focussed on

generating neutralising antibodies to the viral major glycoprotein gp350 that elicits neutralising antibodies during natural infection [14,15]. When tested in a phase II trial, a gp350 vaccine decreased incidence of IM but did not protect against EBV infection [16]. Subsequent work has therefore sought to boost vaccine-mediated protection through two broad strategies. The first is boosting antibody levels, using polymeric presentation systems to assemble soluble gp350 protein into multimeric particles and broadening the range of EBV glycoproteins in the vaccine [17–21]. The second is harnessing T-cell responses to EBV latent cycle proteins, either alone as tested in small phase I trial of an EBNA3A epitope peptide [22] or, in the rhesus lymphocryptovirus model of EBV, in combination with gp350 to elicit anti-viral T-cells alongside neutralising antibodies [23]. In both cases, however, the protective effect of the latent-antigen specific T-cell response appeared limited. These studies highlight the need to identify the EBV-specific T-cell responses most able to recognise and eradicate newly infected cells and thus prevent EBV establishing permanent infection of its host. Such responses are likely to recognise EBV proteins that fall into one of three categories: i) latent cycle proteins expressed earlier than EBNA3A in the infectious cycle, such as EBNA2 and EBNA-LP [24]. ii) lytic cycle proteins expressed in the first 1–2 days after-infection, such as BZLF1, BRLF1, BMLF1 and BMRF1 [24]. iii) Viral structural proteins that are delivered into newly infected cells as preformed virion components. The T-cell subset best able to recognise infected cells, CD8 versus CD4, may also vary for each category of antigen due to differences in antigen accessibility into the MHC-I and MHC-II antigen processing pathways [25].

Harnessing T-cell responses against the EBV structural proteins is an attractive strategy because, unlike T-cells targeting EBV proteins expressed from the viral genome, they can potentially recognise newly-infected cells through antigenic processing of viral proteins brought into cell by the virion [26]. Thus, infected cells can be eliminated before the virus can start the transformation process and start to express its immune evasion proteins to interfere with MHC-I and MHC-II presentation. Proteomic analysis has identified 33 different structural proteins within purified EBV virions which could potentially be used for this vaccine strategy [27]. Currently T-cell epitopes have been mapped to only a small number of these proteins. These are the glycoproteins BLLF1 (gp350), BALF4 (gp110), BXLF2 (gH), BZLF2 (gp42), BKRF2 (gL), BILF2 (gp78) and the tegument protein BNRF1 [2,6,26,28–32]; T-cell responses to the EBV capsid proteins BORF1 and BcLF1 have been detected but not yet characterised [33]. Furthermore, only a small number of T-cells against these epitopes have been investigated for their ability to recognise newly-infected B-cells, which is likely to be essential to limit EBV infection [24]. Therefore, the general rules governing which EBV structural proteins are likely to be most protective against infection are unknown. Our study therefore had three aims: first, discover new T-cell responses to EBV structural proteins and particularly the capsid proteins that have not been studied; second, generate T-cell clones to map and restrict those responses and whether they appear early in infection or during convalescence; third, use T-cell clones to identify which EBV-specific T-cells can recognise B-cells newly-infected with small quantities of virus to mimic recent exposure of a host to the virus.

## Results

### *Ex vivo* T-cell responses to EBV structural antigens

To characterise the total T-cell response to EBV structural proteins in an unbiased manner we constructed a panel of modified vaccina Ankara (MVA) viruses each expressing a single EBV gene: the capsid proteins BcLF1, BORF1, BDLF1, the tegument protein BNRF1 or the viral glycoprotein gp350. Each EBV gene was fused to the invariant chain targeting sequence that, by delivering the resultant fusion protein into the endosomal/lysosomal pathway, produces high

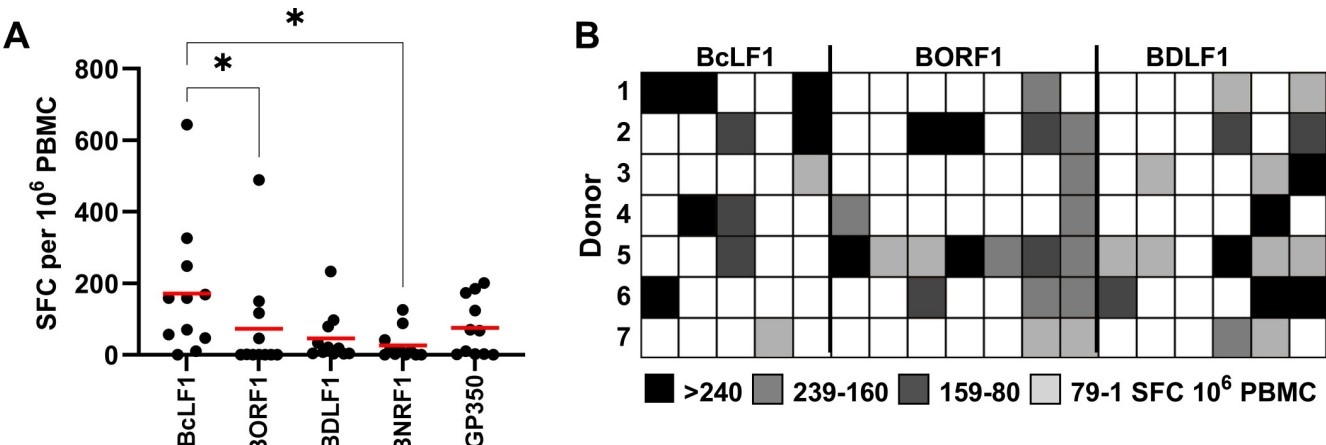

**Fig 1. Characterisation of the frequency of response to EBV structural antigens by ELISpot.** A) The frequency of T-cell responses to EBV structural antigens *ex vivo* by overnight IFNγ ELISpot in 11 healthy donors. Horizontal bar indicates mean response. Significant differences between responses were determined by Friedman test with Dunn's multiple comparison test (*p<0.05). B) Healthy donor PBMC were stimulated with pools of peptides (indicated by columns, each containing 3 peptides) derived from the indicated EBV antigens. Following 7-day culture, response to the peptide pools was assessed by overnight IFNγ ELISpot. Individual rows of the heat map show the response from 7 healthy donors to peptide pools derived from each protein.

levels of MHC-I and MHC-II antigenic peptides thus stimulating CD8+ and CD4+ T-cells [34]. The results of initial *ex vivo* ELISpot screening experiments, infecting PBMCs from eleven EBV-positive donors with each of the above viruses (or a control virus lacking any EBV gene) are shown in **Fig 1A**. The most frequently recognised antigen was BcLF1, with T-cell responses detected in 9/11 donors, followed by gp350 (6/11 donors). Fewer donors possessed responses to BORF1, BDLF1 and BNRF1. The frequency of response to BcLF1 was significantly higher compared to BORF1 and BNRF1 (p = 0.015, Friedman test with Dunn's multiple comparison test). When present, however, T-cell responses to BORF1 were generally comparable in magnitude to those to BcLF1 (mean of responding donors, 161 vs 189 sfc/10$^6$ PBMC, respectively).

T-cell epitopes have previously been mapped to the EBV structural proteins BNRF1 and several glycoproteins [2,6,26,28–32]. By comparison, little is known about the T-cell response to the capsid proteins BcLF1, BORF1 and BDLF1. To investigate the breadth of the T-cell response to the EBV capsid proteins we first performed a series of cultured T-cell assays. Multiple pools of 20-mer peptides, capable of stimulating CD8+ and CD4+ T-cells, were used to stimulate PBMCs from seven donors. Following seven days in culture the PBMCs were tested in ELISpot assays using either the original stimulating peptides or a negative control. This more sensitive approach showed each donor possessed a T-cell response to BcLF1, BORF1 and BDLF1 (**Fig 1B**). Almost every peptide pool elicited a response, showing multiple epitopes exist within each protein; indeed, individual donors often had multiple T-cell responses to the same protein.

## Characterisation of T-clones specific for EBV capsid antigen epitopes

Using limited dilution cloning we successfully established CD8+ and CD4+ T-cell clones specific for BcLF1 and BORF1 but not BDLF1. Each clone was characterised using standardised assays (**S1 Fig**) to identify their cognate epitope peptide and measure their functional avidity (**Fig 2A** and **Table 1**). We observed that CD4+ T-cell clones specific for the HLA-DR15-restricted NIL epitope from BORF1 always responded strongly to the unmanipulated autologous or HLA-matched LCLs carrying the standard B95.8 laboratory strain of EBV (**Fig 2B**). B95.8 LCLs are semi-permissive for virus replication, with typically 1–5% of cells in a culture

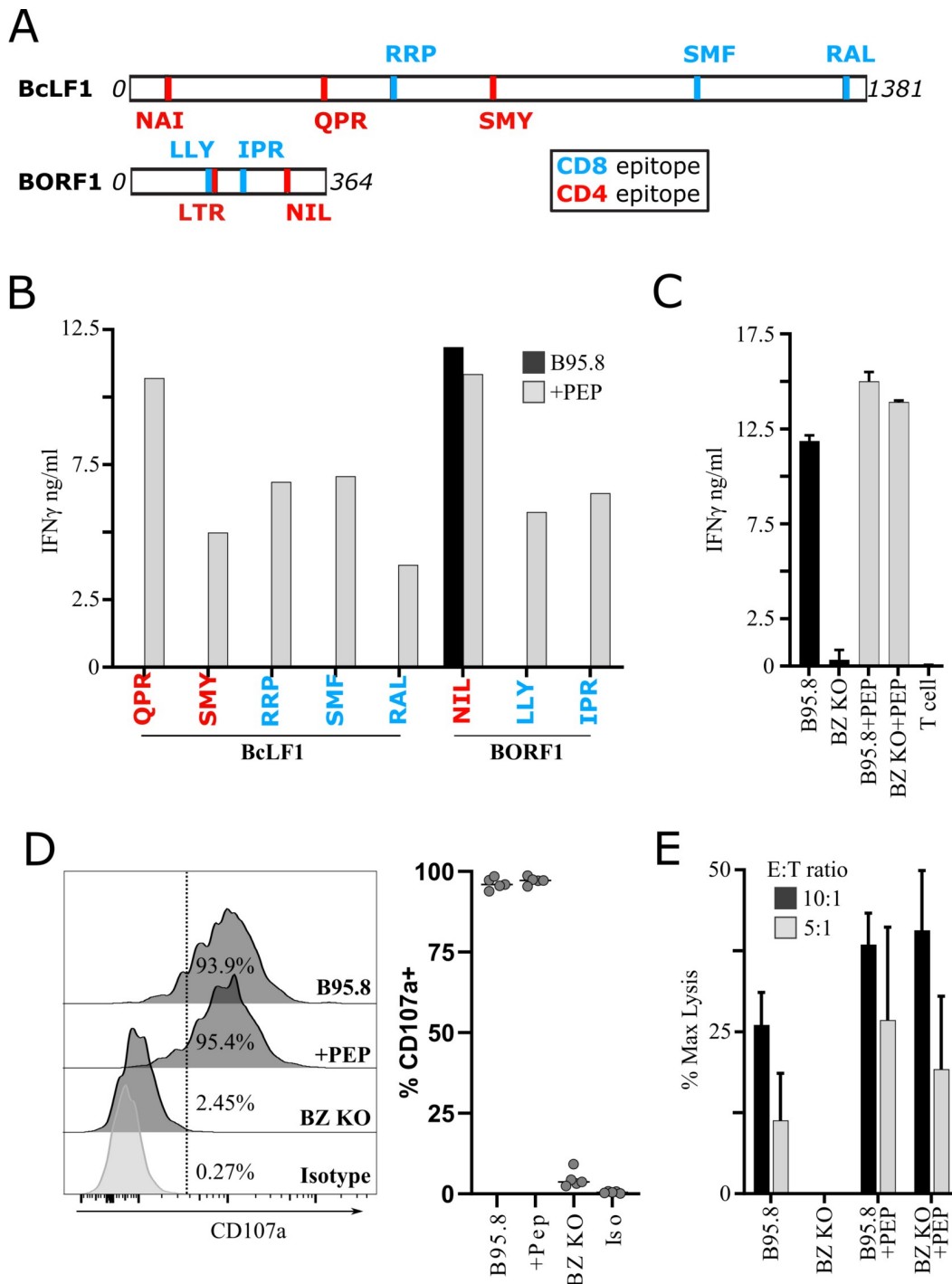

**Fig 2. *In vitro* characterisation of LCL recognition by capsid antigen-specific T-cells.** A) Diagrammatic representation of newly identified epitopes from BcLF1 and BORF1. B-E) Show response of T-cell clones to either unmanipulated B95.8 (semi-permissive for lytic cycle) or BZLF1 deficient (BZ KO, non-permissive) LCL, +PEP indicates target LCLs have been loaded with peptide prior to addition of T-cell clones. B) Clonal T-cells specific for newly defined EBV capsid epitopes were co-cultured o/n with autologous B95.8 LCL which had been either pre-exposed or not to cognate peptide. Responses were measured by IFNγ ELISA on culture supernatant. C-E) BORF1 NIL-specific CD4+ T-cell clones were tested against B95.8 (permissive) or BZLF1-KO (BZ KO, non-permissive) autologous LCLs either pre-pulsed or not with epitope peptide. C) BORF1 NIL-specific clones were co-cultured with B95.8 or BZLF1-KO LCL as indicated and IFNγ release following overnight culture was measured by ELISA. Representative of five BORF1 NIL-specific clones tested, bars indicate mean±SD. D) CD107a degranulation assay showing CD107a expression measured by flow cytometry after 5hrs co-culture in the

presence of anti-CD107a and monensin. A representative example is shown for one clone (left), and results from 5 individual BORF1 NIL-specific T-cell clones (right), bars indicate mean. E) Representative 5hr chromium release assay of BORF1 NIL-specific effectors against the indicated targets performed at 10:1 or 5:1 effector:target ratio. Representative of two independent experiments with different clones, bars indicate mean±SD.

spontaneously entering lytic cycle [35]. We therefore hypothesised that a sub-population of lytic cells may have sensitised the LCL culture for T-cell recognition as observed by ourselves and others [6,26,36]. Accordingly, the NIL-specific CD4+ T-cell clones never responded to unmanipulated BZLF1-K/O LCLs generated from the same donor that, due to deletion of the BZLF1 gene [37], were unable to enter lytic cycle demonstrating recognition is dependent on protein presentation (**Fig 2C**). These BZLF1-K/O LCLs provided a means to characterise the NIL T-cells without the results being obscured by background recognition. Consistent with the interferon-gamma (IFNγ) data, the NIL-specific CD4+ T-cell clones degranulated and displayed CD107a on the cell surface when exposed to unmanipulated autologous or HLA-matched B95.8 LCLs but not their BZLF1-K/O counterparts (**Fig 2D**). Unmanipulated HLA-matched B95.8 LCLs, but not BZLF1-K/O LCLs, were also lysed by NIL-specific CD4+ T-cell clones in $^{51}$Cr cytotoxicity assays (**Fig 2E**).

## EBV capsid antigen-specific CD4+ T-cells, but not CD8+ T-cells, robustly recognise newly infected B-cells

When EBV infects a cell the viral structural proteins, including BcLF1, BORF1 and BNRF1, enter the cell within the incoming virion before any virally-encoded immune evasion or growth transforming genes are expressed [38]. Structural proteins therefore represent a unique viral vulnerability, providing an immediate source of antigen for T-cell recognition of newly infected cells [6,24,26]. To explore this concept in detail, we developed an *in vitro* system to measure T-cell recognition of B-cells infected with small quantities of virus thus mimicking initial exposure of a new host to EBV. In addition to infectious virions, crude viral preparations contain large quantities of soluble viral proteins released from dying virally-infected cells. To remove the confounding effects of the latter, we prepared purified wild-type 2089 EBV virions by sucrose gradient centrifugation and used these to infect B-cells. We first determined the amount of purified EBV virions that would reliably transform purified B-cells in our *in vitro* assays, measuring B-cell infection efficiency 72hrs post-infection using EBNA2 protein expression as a surrogate of transformation (**Fig 3A**). Although we could detect

**Table 1. Newly defined capsid antigen T-cell epitopes.**

| | Protein | Function | Epitope Sequence | Restriction | Functional Avidity (nM) |
|---|---|---|---|---|---|
| **CD4:HLA Class-II** | **BcLF1** | **Major capsid protein** | **SMY**RKIYGEL IALEQALMRL | DRB1*01:01 (DR1) | 40 |
| | | | **QPR**VPISAAVIKLGNHAVAV | DRB1*01:01 (DR1) | 200 |
| | | | **LTR**MANLLYDSPATLADLVP | DRB1*04:01 (DR4) | 800 |
| | | | **NAI**QYVRFLETALAVSCVNT | DRB5*01:01 (DR51) | 800 |
| | **BORF1** | **Capsid triplex subunit 1** | **NIL**RIYYSPSIMHRYAVVQP | DRB1*15:01 (DR15) | 20 |
| **CD8:HLA Class I** | **BcLF1** | **Major capsid protein** | **RAL**IDEFMSV | HLA-A*02:01 (A2) | 4 |
| | | | **SMF**IGTPNV | HLA-A*02:01 (A2) | 0.4 |
| | | | **RRP**NHMNVL | HLA-C*06:02 (Cw6) | 20 |
| | **BORF1** | **Capsid triplex subunit 1** | **LLY**DSPATL | HLA-A*02:01 (A2) | 4000* |
| | | | **IPR**STVKVTV | HLA-A*11:01 (A11) | 20 |

*Titration based on 9-mer derived from peptide prediction sequence.

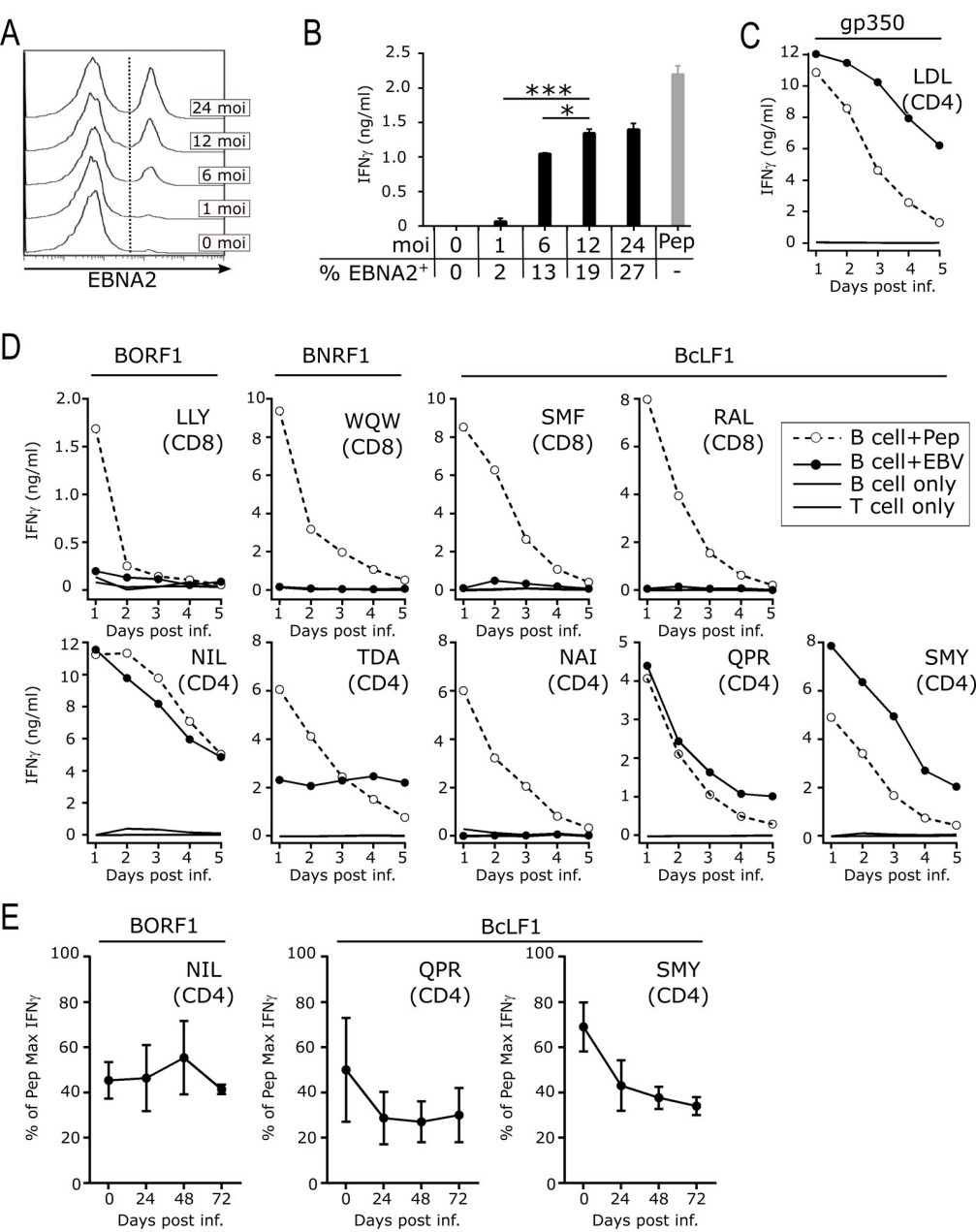

**Fig 3. Recognition of newly infected primary B-cells by EBV capsid antigen specific CD4+ T-cell clones.** A) Representative EBNA2 staining and detection by flow cytometry to assess active infection of isolated primary B-cells infected with increasing moi of WT EBV virus. B) B-cells infected with EBV at indicated moi or peptide pulsed B-cells (Pep) were co-cultured for 24hrs with gp350 LDL-specific CD4+ T-cell clone known to recognise newly infected B-cells. Supernatant was assessed for IFNγ secretion by ELISA. The proportion of infected B-cells, as determined by EBNA2 staining in the same experiment, is indicated for each moi. Repeated measures one-way ANOVA with Geisser-Greenhouse correction and Dunnett's multiple comparison test ($^*p < 0.05$, $^{***}p < 0.001$). C&D) HLA-matched isolated primary B-cells infected with WT EBV at an moi of 12 (filled circle) or pulsed with peptide (open circle, dashed line) were co-cultured with (C) gp350 LDL-specific CD4+ T-cell clone known to recognise newly infected B-cells, or (D) CD4+ and CD8+ T-cell clones specific for BORF1, BNRF1 or BcLF1 epitopes. Supernatant sampled at 24hr intervals as indicated was assessed for IFNγ by ELISA. Clones were tested in two or more independent assays. E) HLA-matched isolated primary B-cells infected with EBV at an moi of 12 were mixed with T-cell clones at the indicated time points following the initial infection. After 24hr co-culture supernatant was harvested and response assessed by IFNγ ELISA, results show mean±SD from three independent experiments. There was no significant difference in IFNγ production by T-cells exposed to newly-infected B-cells (day 0) compared to levels produced after 1, 2 or 3 days had passed since infection occurred ($p > 0.05$, repeated measures one-way ANOVA with Geisser-Greenhouse correction and Dunnett's multiple comparison test).

EBNA2-positive B-cells after infection at a multiplicity of infection (moi) of 6, such a low level of virus was insufficient to give reliable transformation in every experiment. A slightly higher moi (typically 10–12 moi) was therefore used to ensure reliable B-cell infection while keeping the viral input as low as possible. In validation experiments a gp350-specific CD4+ T-cell clone, previously shown to recognise B-cells 24 hours after infection at an moi of 100 [24], still recognised B-cells infected at the much lower moi of 12 (**Fig 3B**). Increasing the moi from 12 to 24 did not significantly increase T-cell recognition but decreasing the moi from 12 to 6 significantly reduced recognition (p = 0.677, p = 0.037, respectively, RM one-way ANOVA with Geisser-Greenhouse correction and Dunnett's multiple comparison test). Furthermore, infection at an moi of 6 occasionally failed to transform B-cells. Thus, a higher moi of 10–12 was used in further experiments.

We then performed a large series of experiments using purified EBV virions to infect purified B-cells from healthy adult donors (of appropriate HLA class I/II type) before coculturing them with various CD8+ and CD4+ T-cell clones. Culture supernatant was sampled 24 hours after infection and every 24 hours thereafter for five days and T-cell recognition quantified by measuring IFNγ in the supernatants by ELISA. As expected, the gp350-specific CD4+ T-cell clone used in the validation experiments responded strongly to B-cells at the earliest timepoint sampled, one day post infection (**Fig 3C**). As for T-cells specific for non-glycoproteins, two previously described CD8+ and CD4+ T-cell clones specific for epitopes within BNRF1 [2] and the newly identified CD4+ T-cell epitope NAI in BcLF1 did not recognise newly-infected B-cells; the T-cells were clearly functional as they produced high levels of IFNγ when exposed to peptide-pulsed B-cells (**Fig 3D**). However, a different pattern of results was seen for the other capsid-specific T-cell clones. Although newly infected B-cells were not recognised by BORF1 or BcLF1-specific CD8+ T-cell clones, CD4+ T-cell clones specific for the BORF1 NIL and BcLF1 QPR or SMY epitopes rapidly produced high quantities of IFNγ (**Fig 3D**).

The fact that EBV capsid proteins within infectious virions can sensitise newly infected B-cells for CD4+ T-cell recognition raised the important question of how long the B-cells would remain visible to T-cell surveillance. The above experimental design, repeatedly sampling culture media from the B-cell + T-cell cocultures established at the start of the time course, was unable to determine whether T-cell epitope display on the surface of newly infected B-cells was transient or was sustained over time. We therefore performed an additional series of experiments, infecting B-cells at the start of the experiment as before but now adding T-cells 0, 1, 2 or 3 days after infection. Culture supernatants were then harvested 24 hours after T-cell addition and IFNγ measured by ELISA. Thus, the T-cells were exposed to the B-cells for the same 24 hour period but after different lengths of time had passed since initial infection occurred. As shown in **Fig 3E**, recognition of the infected B-cells by the BORF1- and BcLF1-specific CD4+ T-cell clones was not significantly different between day 0 and days 1, 2 or 3 post infection (p>0.05, RM one-way ANOVA with Geisser-Greenhouse correction and Dunnett's multiple comparison test). The ability of the BORF1- and BcLF1-specific CD4+ T-cell clones to recognise B-cells up to three days after infection showed the cognate epitopes were displayed on the infected B-cell surface for long periods of time after infection, as previously reported for MHC-II epitopes from other EBV proteins [25,36].

## BORF1-specific NIL CD4+ T-cells control B-cell transformation

Having shown EBV capsid-specific CD4+ T-cells can recognise newly-infected B-cells, we investigated whether they could prevent B-cell transformation by EBV. A fixed number of purified B-cells, infected with purified EBV virions as before, were added to 96-well plates and cultured alone or with various numbers of T-cells at effector:target (i.e. T-cell:B-cell) ratios

ranging from 0:1 (no T-cells added) to 1:1 (equal numbers of T-cells and B-cells). Cultures were scored visually for B-cell outgrowth after 21–28 days. In each experiment the B-cell donor was carefully selected to be HLA-matched to T-cell clones specific for multiple capsid antigen epitopes. Illustrative examples from two experiments are shown in **Fig 4A**. In some cases the results in the outgrowth assays were consistent with the earlier experiments that used IFNγ as the readout. Thus, the control gp350 LDL-specific CD4+ T-cell clone efficiently blocked B-cell outgrowth whereas no CD8+ T-cell clone tested ever prevented B-cell outgrowth even at the highest E:T ratio (**Fig 4A**). Likewise, the BcLF1 NAI- and BNRF1 TDA-specific CD4+ T-cell clones, both of which failed to recognise LCLs and newly infected B-cells, also failed to prevent outgrowth. However, results for the BcLF1 QPR- and BcLF1 SMY-specific CD4+ T-cell clones were not concordant; although both had efficiently recognised newly infected B-cells in the earlier experiments measuring IFNγ production, both failed to prevent B-cell outgrowth in these longer assays. Importantly, however, multiple BORF1 NIL specific CD4+ clones, all of which had efficiently recognised LCLs and newly infected B-cells in the earlier experiments, strongly inhibited B-cell outgrowth (**Fig 4A and 4B**). T-Cell Receptor (TCR) sequencing confirmed these four clones had distinct V-beta T-cell receptor sequences, as such these clones originated from distinct T-cell clonotypes (**S1 Table**) demonstrating this to be a common function of BORF1 NIL specific CD4+ T-cells.

Cytotoxicity is essential for T-cells to control the outgrowth of EBV-transformed LCLs *in vitro* [39,40]. We therefore examined whether differences in cytotoxic potential might explain why, despite similar results in the IFNγ experiments, the NIL-specific CD4+ T-cells efficiently controlled B-cell outgrowth while the BcLF1 NAI- SMY- and QPR-specific CD4+ T-cells did not. Using flow cytometry, we detected high levels of intracellular perforin and granzyme B in CD8+ T-cell clones (**Figs 4C and S2**). In agreement with previous studies on latent antigen-specific CD4+ T-cell clones [40], we also detected perforin and granzyme B in the CD4+ T-cell clones specific for the EBV lytic antigens although levels varied between clones. Notably, levels of these cytotoxic effector molecules were high in the BORF1 NIL-specific CD4+ T-cell clones that had controlled outgrowth but low in the SMY-, and QPR-specific CD4+ T-cell clones that produced IFNγ when exposed to infected B-cells but had failed to control their outgrowth. Indeed, the level of perforin was significantly higher in BORF1 NIL clones than other CD4 clones, but not significantly different to CD8 clones (p = 0.018, p = 0.847; NIL vs CD4 and CD8 clones respectively, one-way ANOVA with Tukey's multiple comparison test). Finally, repeating the outgrowth assay with B-cells infected at a much higher moi of 100 showed that an NIL-specific CD4+ T-cell clone could still control outgrowth under this more challenging condition, albeit with slightly reduced efficacy. Growth inhibition was also unaffected by cyclosporin A, which blocks T-cell cytokine production but not cytotoxicity [40,41] clearly showing cytotoxicity is essential for CD4+ T-cell control of newly-infected B-cells.

## *Ex vivo* characterisation of BORF1-specific NIL CD4+ T-cell responses during primary EBV infection

Circulating CD4+ T-cell responses to most EBV lytic and latent antigens can be detected soon after infection while EBNA1 specific responses are delayed, taking up to six months to develop [5]. To explore the dynamics of BORF1 NIL-specific CD4+ T-cell response in newly infected people we used MHC-II tetramers to test PBMCs from six people with IM who had recently been infected by EBV. All six people were HLA-DR15 positive and thus had the potential to mount T-cell responses to the BORF1 NIL epitope. Some of these donors were also positive for HLA-DR1 allowing responses to the BcLF1 SMY and QPR epitopes to be examined in parallel. An example of tetramer staining, gating strategy and validation of tetramer staining is

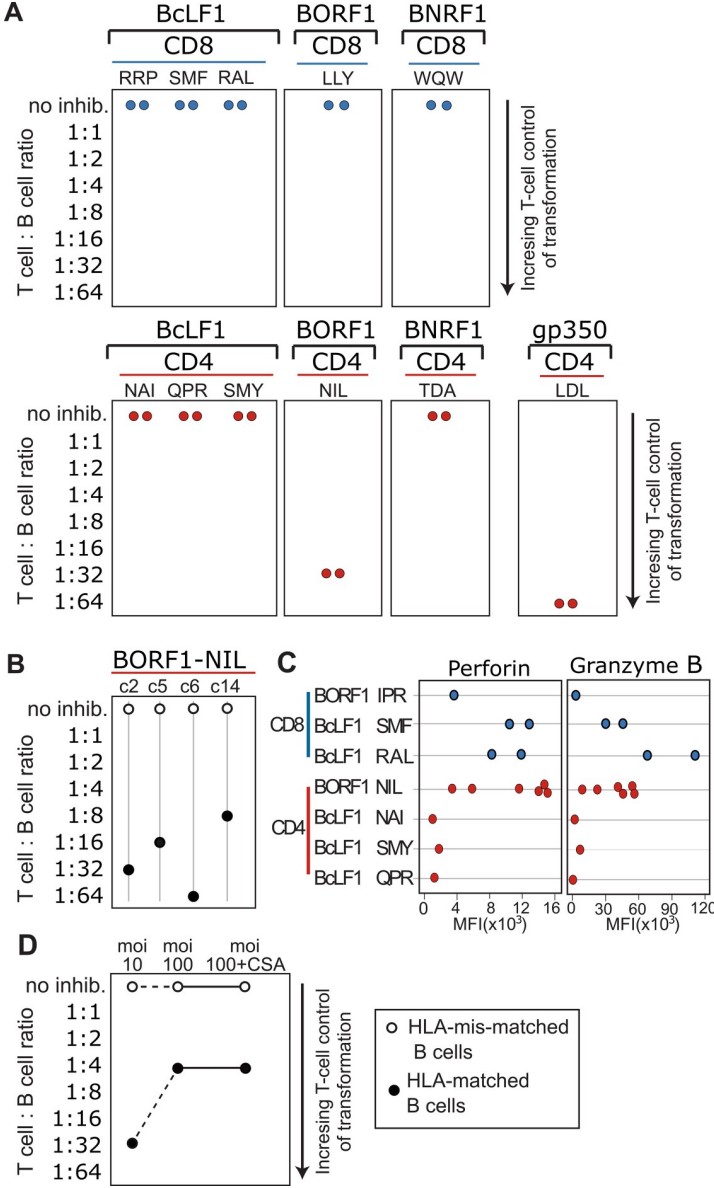

**Fig 4. Control of newly EBV infected B-cell outgrowth by EBV capsid antigen-specific T-cells.** Isolated primary B-cells were infected with WT EBV at an moi of 12 and incubated in an equal volume with decreasing number of T-cell clones as shown, in triplicate. Outgrowth of B-cells was visually assessed at 21–28 days after infection. A) Indicates the ratio of T-cells to B-cells required to prevent outgrowth of transformed B-cells. B) Shows outgrowth assays using four further NIL reactive CD4+ T-cell clones with HLA-matched (black circles) or mis-matched (white circles) B-cells. C) Perforin and granzyme B expression was assessed by intracellular staining of the indicated CD8+ and CD4+ T-cell clones, each point represents an individual T-cell clone. Results are presented as mean fluorescent intensity (MFI) of the T-cell clone as assessed by flow cytometry. Comparing the BORF1 NIL-specific CD4+ T-cell clones to other CD4 + and CD8+ T-cell clones, the levels of perforin were significantly higher in NIL clones compared to other CD4+ but not CD8+ clones (p = 0.018, p = 0.847; respectively, one-way ANOVA with Tukey's multiple comparison test). D) HLA-matched (black circles) or mis-matched (white circles) primary B-cells were infected at an moi of 10 or 100 before co-culture with NIL-specific CD4+ T-cell clone. Cyclosporin A was added at 1ug/ml where indicated.

provided in **S3 Fig**. As a comparator we also measured responses to a CD4+ T-cell epitope within the EBV latent cycle antigen EBNA2, HLA-DR52b restricted PRS, that is immunodominant during IM [5]. All six donors had NIL-specific CD4+ T-cell responses at frequencies

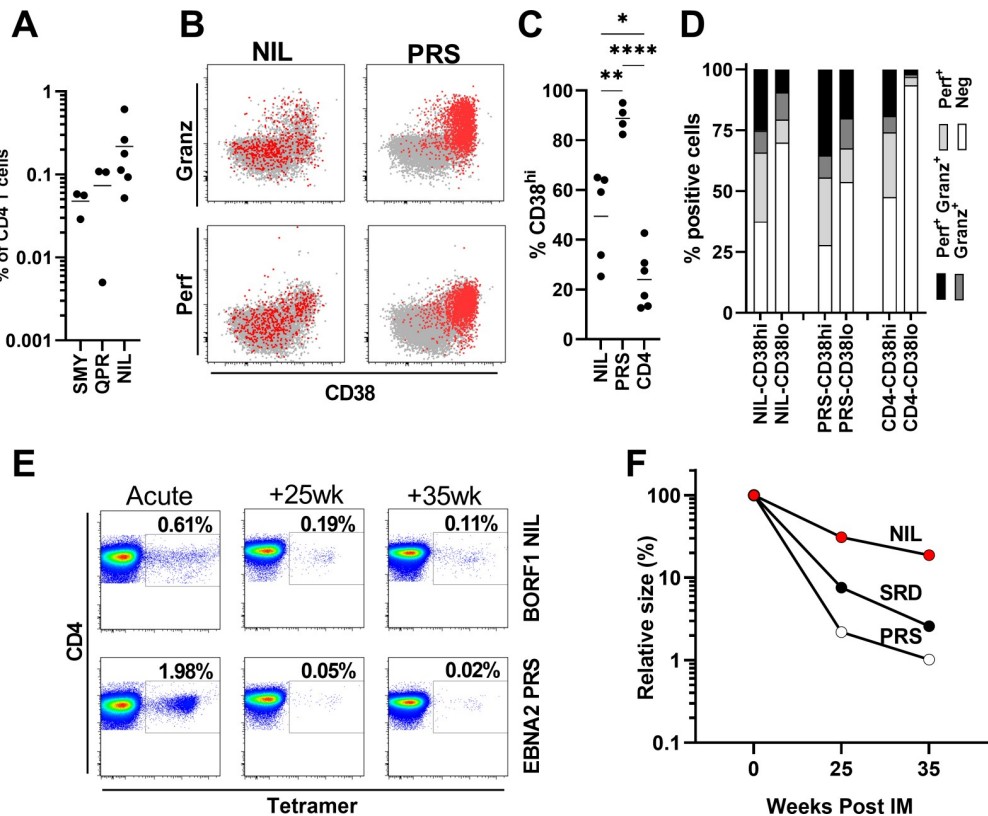

**Fig 5. Characterisation of capsid-specific CD4+ T-cell responses in acute EBV infection and convalescence using MHC-II tetramers.** A) Frequency of MHC-II tetramer-specific CD4+ T-cells in six acute IM patients. Note only three donors had appropriate HLA-type for BcLF1 tetramer staining. Bars indicate mean. B) Representative CD38 and perforin or granzyme B expression in BORF1-NIL or EBNA2-PRS tetramer-positive CD4+ T-cells (red), or bulk CD4 + T-cells (grey) in acute IM. C) Summary of the frequency of CD38hi NIL tetramer-positive CD4+ T-cells, PRS tetramer-positive CD4+ T-cells, or bulk CD4+ T-cells. Dots indicate individual IM donors, bar indicates mean. Results of one-way ANOVA with Tukey's multiple comparison test are shown (*p<0.05, **p<0.01, ****p<0.0001). D) Summary of perforin and granzyme B expression by CD38hi and CD38lo/- cells. NIL tetramer-positive CD4+ T-cells, PRS tetramer-positive CD4+ T-cells, or bulk CD4+ T-cells were assessed by intracellular flow cytometry of blood samples from acute IM donors. E) Representative example from one of two IM donors showing the frequency of MHC-II tetramer-positive cells in acute IM through to convalescence. F) Graph showing the decline of BORF1-NIL, EBNA2-PRS and BaRF1-SRD specific CD4+ T-cell populations from acute IM into convalescence. Results are expressed as a percentage of each CD4+ T-cell population's frequency in acute IM.

between 0.04 to 0.6% of circulating T-cells (**Fig 5A**) in line with frequencies detected by earlier work on other EBV antigens during IM [5]. Based on CD38 expression (**Fig 5B and 5C**) 24–63% of NIL-specific T-cells were activated, a significantly higher frequency than the bulk CD4+ T-cell population in the same individuals but lower than the EBNA2-PRS specific CD4+ T-cells that are known to be highly activated in IM (p = 0.022, p = 0.002, respectively, one-way ANOVA with Tukey's multiple comparison test) [5,42]. Perforin and granzyme B expression by EBV-specific CD4+ T-cells is high during IM and associated with T-cell activation [42,43]. Compared to PRS-specific CD4+ T-cells, levels of perforin and granzyme B were overall lower in the NIL-specific T-cells (**Fig 5C**). However, this likely reflects the lower proportion of activated cells in the latter as, when only CD38-positive activated T-cells were examined, levels of perforin and granzyme B were similar (**Fig 5D**).

As IM patients recover the frequency of EBV-specific CD4+ T-cells in their blood decrease to lower but stable levels [5]. To examine the dynamics of the BORF1 NIL response over time

we obtained convalescent samples from two of the IM patients. An illustrative example from one donor is shown in **Fig 5E**. As expected from previous work [5,42] the EBNA2 PRS response rapidly contracted over time, a peak response of 1.98% of CD4+ T-cells during IM decreasing to 0.05% six months and 0.02% eight months after IM. Although the peak frequency of BORF1 NIL-specific CD4+ T-cells was lower than the coincident PRS response during IM, over time it contracted more slowly to 0.19% of CD4+ T-cells 25 weeks and 0.11% 35 weeks after IM. This contraction of NIL-specific CD4+ T-cells was also slower than another CD4+ T-cells response we also examined in this patient, the HLA-DR7 restricted epitope SRD from BaRF1 (**Fig 5F**). Examining a second donor, we observed the same slow contraction of the NIL-specific T-cell response over time; this donor's HLA type, however, prevented us from examining any control T-cell responses (**S4 Fig**).

### *Ex vivo* characterisation of BORF1-specific NIL CD4+ T-cells in persistently EBV infected donors

We next used the HLA-II tetramers to examine the BORF1 NIL-specific CD4+ T-cell response in healthy donors carrying EBV long-term as a persistent infection. By choosing donors that were also positive for HLA-DR1 we were also able to examine T-cell responses to the BcLF1 CD4+ T-cell epitopes SMY and QPR in parallel. Example staining from one donor is shown in **Fig 6A**. NIL-specific CD4+ T-cells were readily detectable in all donors at frequencies ranging from 0.010–0.323% of CD4+ T-cells (**Fig 6B**). SMY- and QPR-specific CD4+ T-cells were also present but were always much lower in frequency (0.005–0.001%) and often close to the limit of tetramer detection.

Examining the phenotype of NIL-specific CD4+ T-cells in three patients with sufficiently large populations of these cells (**Fig 6C and 6D**) showed they were predominantly CD45RA⁻ CCR7⁻ effector memory T-cells (Tem, 16.9% versus 93.7% for total CD4+ and NIL-specific T-cells respectively). Almost all (90.9%) NIL-specific CD4+ T-cells expressed granzyme-B and almost half (45.8%). expressed perforin An illustrative example for one donor is shown (**Fig 6E**) and proportions for individual donors are also presented (**Fig 6F**). In contrast to EBNA-2 PRS-specific CD4+ T-cells during latent infection [42], perforin and granzyme B expression by NIL-specific CD4+ T-cells was in the absence of continued activation as shown by their lack of CD38 expression (**Fig 6G and 6H**).

### *Ex vivo* analysis of BORF1-specific CD4+ T-cell function in persistent EBV infected donors

In a final series of *ex vivo* experiments we utilised the MVA viruses originally used to identify capsid-antigen specific T-cell responses (**Fig 1**) to characterise the functional profile of the cognate T-cell response. Using blood samples from seven persistently EBV-infected donors, B-cells were purified from PBMCs and infected with MVA viruses expressing invariant chain targeted BORF1, BcLF1 or ovalbumin as a control. The B-cells were washed and then added back to the same donor's PBMC preparation. After 5 hours T-cell function was assessed by flow cytometry to detect IFNγ production, as in the earlier ELISpot assays, but now also CD107a surface display to detect degranulation and cytotoxic potential (**Fig 7A**). Across the donors (**Fig 7B**) the response to BcLF1 was dominated by CD8+ T-cells which, as expected, produced IFNγ and de-granulated. The CD8+ T-cell response to BcLF1 was significantly higher than the CD4+ T-cell response (p = 0.023, Friedman test with Dunn's multiple comparison correction). In contrast, BORF1 elicited a more balanced response composed of CD8+ and CD4+ T-cells (p = 0.723, Friedman test with Dunn's multiple comparison). Functionally, the BORF1-specific CD4+ response resembled the BcLF1-specific CD8+ response, with the majority of responding

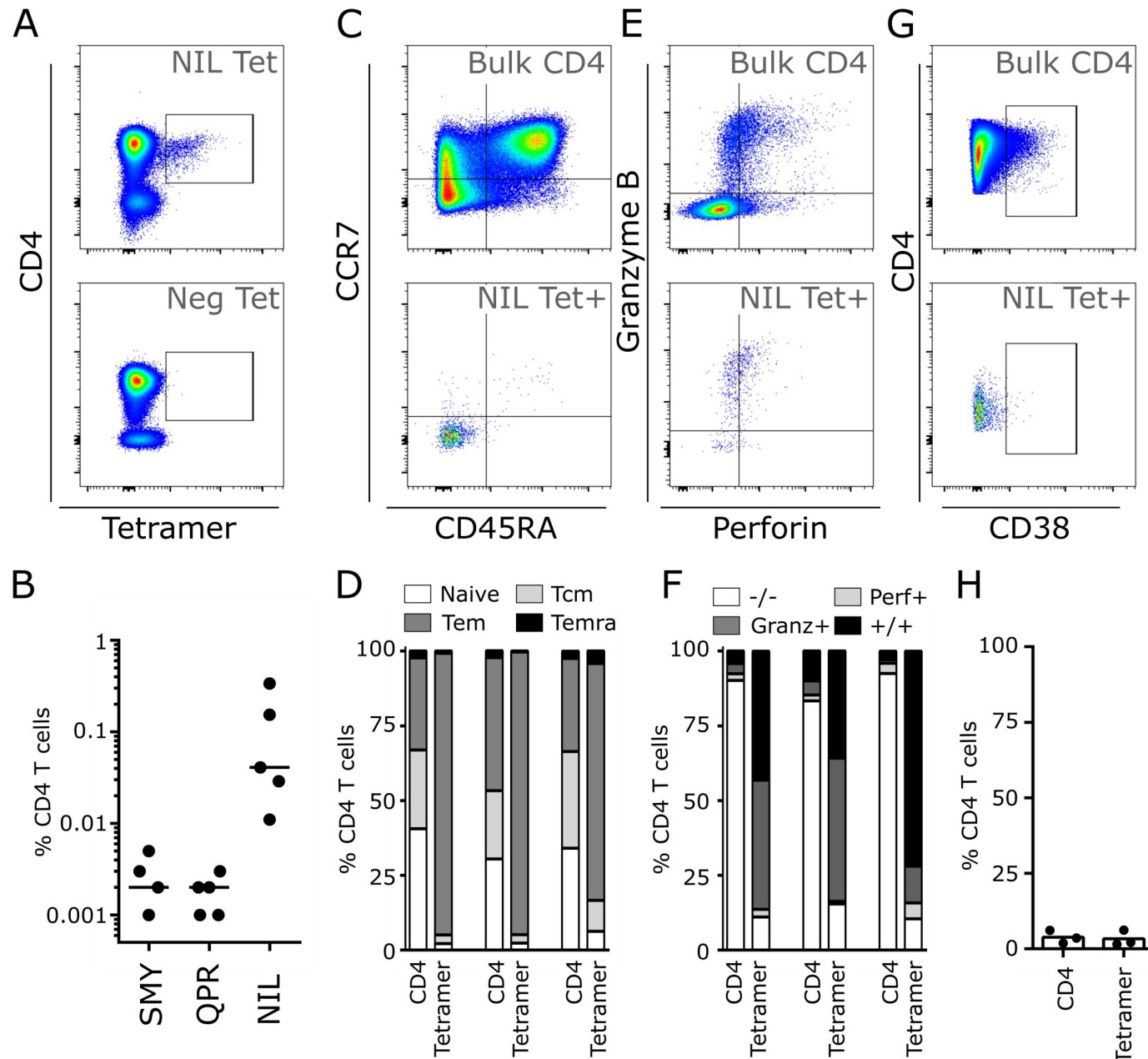

**Fig 6. MHC-II tetramer characterisation of BORF1 specific CD4+ cells in long term convalescent healthy donors.** A) Example flow cytometry staining of total T-cells with either an MHC-II tetramer containing an irrelevant peptide, or newly defined EBV capsid epitope, BORF1 NIL. B) Frequency of tetramer positive cells in five healthy EBV positive donors against the indicated epitopes. Bars indicate mean. C) Example of memory phenotype characterisation of BORF1 NIL tetramer-specific T-cells, compared to the Bulk CD4+ T-cell population. D) Memory phenotype of total CD4+ and BORF1-NIL tetramer positive T-cells in three healthy donors. E) Example perforin and granzyme B staining of BORF1 NIL tetramer-specific T-cells, compared to the Bulk CD4+ T-cell population. F) Perforin and granzyme B expression in total CD4+ and BORF1-NIL tetramer positive T-cells from three healthy donors. G) Example of staining for the late activation marker CD38 on total CD4+ and BORF1 NIL tetramer positive T-cells in one donor. H) Percentage CD38hi total CD4+ and BORF1-NIL tetramer positive T-cells in three donors, bars indicate mean.

cells producing IFNγ and degranulating (**Fig 7A and 7B**). Polyfunctionality, the ability of T-cells to secrete more than one cytokine, correlates with a more effective T-cell response [44–46]. Examining the cytokine profile of the capsid specific T-cells (**Fig 7C**) we found the majority of the BcLF1 CD8+ and BORF1 CD4+ responses were polyfunctional. In contrast, most

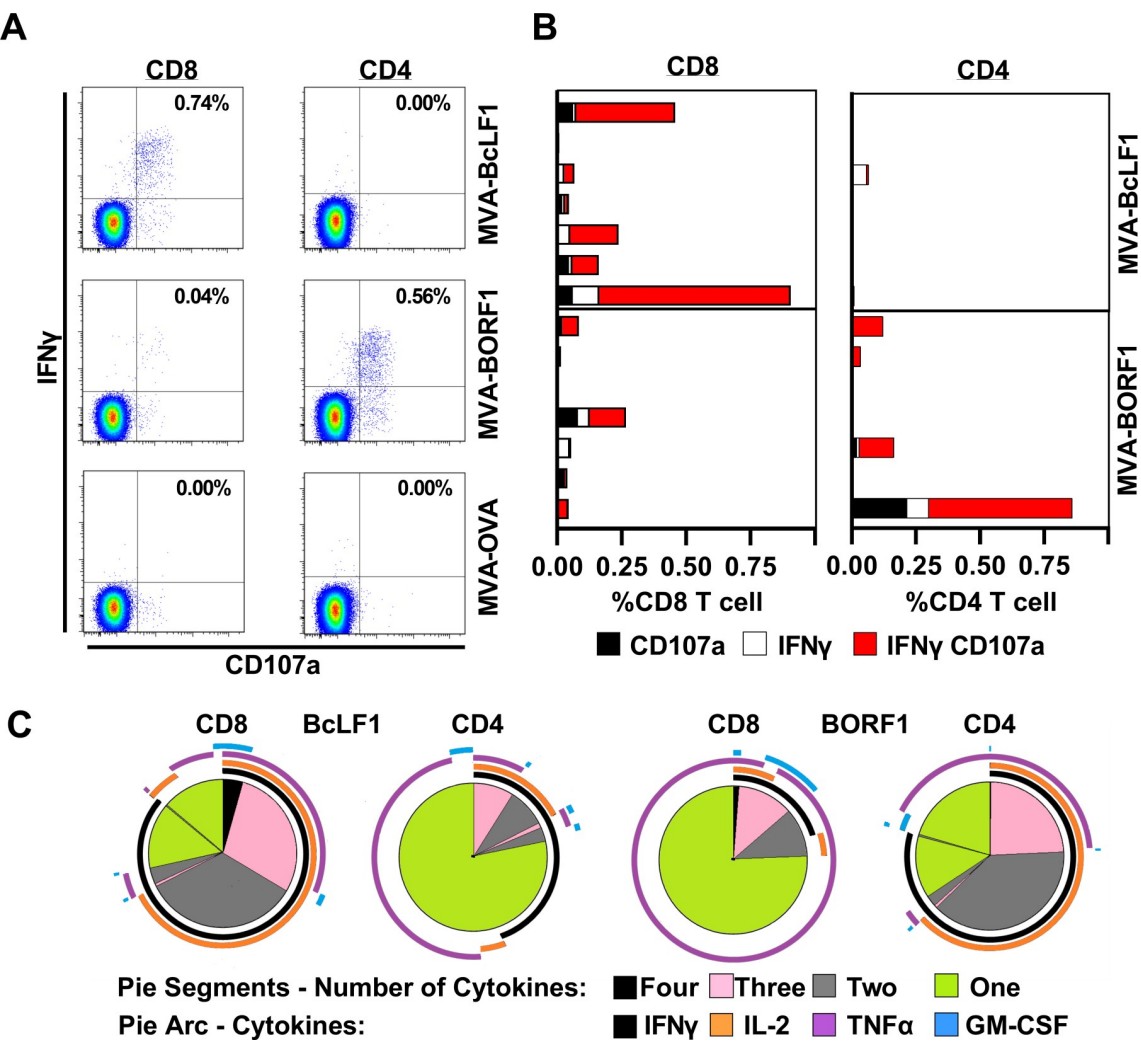

**Fig 7.** *Ex Vivo* **functional characterisation of the T-cell response to naturally processed EBV capsid antigens.** Autologous B-cells were isolated and pre-infected with MVA expressing EBV capsid antigen or ovalbumin (OVA) as a control. Panels A & B) T-cell response in PBMCs was assessed by measuring CD107a degranulation and intracellular IFNγ production by flow cytometry, following 5hr co-culture. A) Example flow cytometry staining from one healthy donor, showing CD8+ BcLF1, and CD4+ BORF1 responses. B) The breath of CD8+ and CD4+ responses to capsid antigens in seven healthy donors; bars represent individual donors. Across all donors, the CD8+ and CD4+ T-cell response to BcLF1, but not BORF1, was significantly different in frequency (p = 0.023 and p = 0.723 respectively, Friedman test with Dunn's multiple comparison correction). C) Spice plots illustrating the cytokine expression profiles of CD8+ and CD4+ T-cells responding to MVA-infected B-cells expressing BcLF1 and BORF1. Data are the mean of 3 healthy donors. Each pie chart segment reflects the fraction of cells expressing the number of cytokines indicated in the key. Arcs represent the distribution of each individual cytokine.

BcLF1 CD4+ and BORF1 CD8+ T-cells produced only a single cytokine. Collectively, these data show that the BORF1-specific T-cell response comprises polyfunctional CD4+ T-cells with cytotoxic potential that are maintained in the absence of activation in healthy latently EBV infected donors.

## Discussion

In newly infected naïve hosts EBV initially replicates in the oropharyngeal epithelium. Persistent infection, the state from which all EBV-associated diseases are thought to arise, is achieved by viral colonisation of the B-cell system. This requires expansion of newly infected B-cells

that then enter latency in the memory B-cell pool, downregulating viral gene expression to evade T-cell surveillance. T-cells able to recognise and eliminate newly infected B-cells are therefore likely to be the most effective in preventing infection. The viral structural proteins present within the incoming EBV virion have the potential to render newly infected B-cells visible to T-cells before the viral transforming and immune evasion genes are expressed. The EBV virion, containing over thirty proteins, is potentially a rich source of T-cell epitopes that could be harnessed for vaccination. Currently only a small number of these proteins have been studied as potential T-cell targets and T-cell epitopes have only been mapped within some of the viral glycoproteins and a single tegument protein, BNRF1.

Previous screening assay results indicated T-cell responses against the EBV viral capsid proteins may also exist [33]. Seeking to broaden the categories of structural proteins that have been studied as potential T-cell targets, we tested three capsid proteins (BcLF1, BORF1 and BDLF1) and showed T-cell responses against all three proteins are common in healthy EBV-positive donors and each protein contains multiple T-cell epitopes. With the caveat that we tested a relatively small number of donors, every individual had a T-cell response to at least one epitope within each of the three proteins; indeed, most individuals possessed multiple responses to each antigen. To characterise these responses in more detail we established T-cell clones against epitopes within two of these proteins, BcLF1 and BORF1, and showed these proteins were targets of CD8+ and CD4+ T-cells. We then examined which of these T-cell responses are best able to recognise EBV infected cells. Using the CD8+ T-cell clones, we found none were able to efficiently recognise EBV transformed LCLs or B-cells exposed to purified virions. These results are consistent with previous studies that showed the main source of CD8+ T-cell epitopes in LCLs and B-cells is intracellularly synthesised protein processed via the normal endogenous antigen processing pathway [25]. Although a proportion of B95.8 transformed LCLs enter lytic cycle, and thus express the structural proteins intracellularly, only epitopes from the immediate early and early lytic cycle proteins are efficiently presented to CD8+ T-cells, as multiple viral proteins act to inhibit MHC-I presentation later in the viral lytic cycle when the structural proteins are expressed [47,48]. Results using the CD4+ T-cell clones were more complex. CD4+ T-cell clones specific for the BORF1 NIL epitope were clearly efficient at recognising LCLs as well as B-cells infected with EBV. However, CD4+ T-cell clones specific for the SMY and QPR epitopes in BcLF1 failed to recognise LCLs but did recognise B-cells that had been exposed to low quantities of EBV virions. These differences in epitope display between LCLs and B-cells could reflect differences in their endosomal protease composition. LMP1 expression in LCLs induces the expression of matrix metalloproteinases (MMPs) that can then potentially access endosomes via endocytosis [49,50]. Cathepsin G activity has also been shown to be higher in LCLs than B-cells leading to increased degradation and reduced presentation of a T-cell epitope in the former [51]. Alterations in endosomal protease composition can alter the CD4+ T-cell epitope repertoire in two ways. Epitopes may be destroyed if they contain cleavage sites, an effect we have previously observed for epitopes within EBNA1 [52]. Alternatively, epitopes may be created if the series of cleavage events that convert an intact protein into epitope peptides is altered [53]. Future work characterising the antigen processing steps by which these epitopes are generated will allow these two possibilities to be separated. These processing differences are likely to also contribute to differences in CD8+ T-cell recognition which have been reported [54]. In these assays, semi-permissive B95.8 LCL exposed to high levels of inactivated virus were recognised by BcLF1 CD8+ T-cell clones, in contrast to our results. These assays are conceptually different from the recognition of newly infected primary B-cells which we present here. The key result, however, was that a limited dose of viral particles were sufficient to sensitise newly EBV-infected B-cells for CD4+ T-cell recognition.

Cytotoxicity is an essential requirement for T-cells to control the transformation of B-cells by EBV. We therefore performed outgrowth assays and showed that CD4+ T-cell clones that were cytotoxic were also able to effectively control B-cell outgrowth. In these experiments we deliberately chose to use a low level of virus, and therefore a low level of antigen, to mimic the situation when a naïve donor might be initially exposed to EBV infection. We also showed that B-cells exposed to higher levels of virus could also be controlled by cytotoxic CD4+ T-cell clones specific for the BORF1 protein. An encouraging result was the longevity of the CD4+ T-cell epitopes on the B-cell surface following EBV exposure, which increases the likelihood of a newly infected B-cell encountering an antigen-specific CD4+ T-cell and being eliminated before the virus can complete cellular transformation.

Our experience, and that of others, is that EBV-specific CD4+ T-cells grown in long-term culture are almost invariably cytotoxic [6,26,30,39,40,55–57]. This is likely due to the repeated stimulations needed to establish and maintain them *in vitro*. A key question, therefore, was whether the capsid protein specific CD4+ T-cells are cytotoxic *in vivo*. Previous work using MHC-II tetramers to characterise EBV-specific CD4+ T-cells directly from the blood, thus avoiding any skewing of their phenotype, has shown the T-cells express perforin and granzyme during IM, when they are activated, but lose expression over time as the patients recover and T-cell activation declines [46]. Using tetramers to detect and phenotype BORF1 NIL-specific CD4+ T-cells we found that they similarly express perforin and granzyme-B during IM and, based on their expression of the T-cell activation marker CD38, had recently encountered antigen. However, BORF1 NIL-specific CD4+ T-cells also expressed perforin and granzyme B in healthy long-term carriers of the virus with no recent history of IM. This result was consistent for all three donors who possessed NIL responses large enough for phenotype assessment. Cytotoxic CD4+ T-cells have mostly been described in situations of chronic antigen exposure such as persistent viral infection and cancer [58–61]. For EBV, based on measurements of viral load in the blood and oropharynx, high antigen load occurs during IM and then for approximately six months afterwards in the throat before declining to the low levels seen in long-term carriers. Thus, outside of the initial infection, antigen supply is limited with only very small numbers of lytically infected cells present in the oropharynx of long-term carriers [62,63]. In this situation we speculate that there may be an important role for antigen presentation by the much larger number of neighbouring cells exposed to proteins released from the sporadic lytically-infected cells. B-cells, which are abundant in these tissues and express MHC-II, are likely to be important stimulators of CD4+ T-cells in this context. Because B-cells have low levels of endocytosis, soluble proteins, such as the latent and early lytic EBV proteins, will be less well presented than structural EBV proteins contained within virions that can efficiently bind to B-cells and be endocytosed. Thus, a potential explanation for the higher levels of perforin and granzyme in the structural antigen-specific CD4+ T-cells could simply be that they are more frequently exposed to antigen stimulation in long-term virus carriers during periodic virus reactivations compared to CD4+ T-cells specific for the non-structural antigens. An alternative explanation could be that a more cytotoxic phenotype is established in structural protein specific CD4+ T-cells during IM and maintained over time. There is increasing interest by viral and tumour immunologists alike in harnessing cytotoxic CD4+ T-cells for therapy [58,61]. Identifying the rules that govern why long-term cytotoxicity is maintained in some but not other CD4+ T-cells is therefore of considerable interest [64].

Because we were unable to source tetramers for the other structural protein epitopes we examined the generalisability of our findings by stimulating PBMCs from long-term EBV carriers, using autologous B-cells expressing the BcLF1 or BORF1 proteins fused to MHC-II targeting tags. CD4+ T-cells specific for both capsid proteins degranulated following antigenic stimulation *ex vivo* consistent with them having a cytotoxic phenotype. This experiment also

allowed us to assess the cytokines these T-cells produced upon activation. In addition to IFNγ, which was used to detect the responses in the original ELISpot assays, the CD4+ T-cells produced IL2, TNF-α and GM-CSF consistent with a Th1 profile. Furthermore, the majority of BORF1-specific CD4+ T-cells and a smaller proportion of BcLF1-specific CD4+ T-cells were polyfunctional, expressing multiple cytokines, consistent with more effective control of chronic viral infection [46,65–67].

The fact that NIL-specific CD4+ T-cells efficiently recognise newly infected B-cells, are present and activated during infectious mononucleosis yet are unable to prevent the virus establishing permanent infection of the host raises the question of whether they would ever be protective. We suggest that the inevitable time lag between initial exposure to EBV and the development of a novel antigen-specific T-cell response in a virus-naïve individual allows the early stages of primary infection to proceed unchecked. However, prior induction (by vaccination) of T-cells specific for antigens that are important early in EBV infection could limit primary infection through rapid recognition of virus-infected B-cells. For CD8+ T-cells, candidate antigens are likely to be those proteins expressed first from the virus genome and processed through the endogenous route, such as EBNA2 and EBNA-LP [24]. In contrast, processing of viral structural proteins, such as the capsid proteins, through the exogenous antigen pathway may provide a window of opportunity for early CD4+ T-cell recognition of the incoming virion before any virally encoded genes are expressed. Together, our data expand the range of EBV-specific T-cell responses and identify new targets for prophylactic EBV vaccine development.

## Material and methods

### Ethics statement

The study was approved by the West Midlands (Black Country) Research Ethics Committee (study reference 14/WM/1254) and all donors provided written informed consent.

### Blood donors, lymphoblastoid cell lines and purified B-cell preparation

Blood samples were collected from donors and peripheral blood mononuclear cells (PBMCs) were isolated by Ficoll-Hypaque centrifugation. Lymphoblastoid cell lines (LCLs) were generated by culturing PBMCs in standard LCL culture media (RPMI 1640 medium with 100 IU/ml penicillin, 100 mg/ml streptomycin, L-glutamine and 10% foetal calf serum) supplemented with 0.1ug/ml cyclosporin A (CSA; Sandimmune; Novartis Pharmaceuticals) with the addition of B95.8 strain EBV or a recombinant EBV lacking the BZLF1 gene (to generate B95.8 or BZLF1-K/O LCLs respectively). B-cells were isolated from PBMC immediately before use in experiments using anti-CD19 Dynabeads, and CD19 Detach-a-beads (Invitrogen) following the manufacturer's instructions. B-cell purity was assessed by flow cytometry and purified B-cell preparations typically contained at least 95% B-cells.

### Production of recombinant Modified Vaccina Ankara viruses

Six gene constructs were designed to encode the following (N to C terminus): an endo/lysosomal targeting sequence composed of the first 80 amino acids of the invariant chain, an inefficient 2A self-cleaving linker (21aamut) [68], the gene of interest, and a nine amino acid influenza hemagglutinin tag. The use of an inefficient 2A tag ensured a portion of the *de novo* expressed protein was targeted to the MHC-II antigen processing pathway within the infected cell with the remainder available for processing by the MHC-I pathway. EBV gene sequences inserted into the construct were derived from B95.8 strain of EBV: BcLF1 (NC_007605.1),

BORF1 (NC_007605.1), BDLF1 (NC_007605.1), BNRF1 (NC_007605.1), GP350 (NC_007605.1). Ovalbumin (NC_052533.1) was also cloned into the construct to serve as a negative control. Synthetic DNA sequences were prepared by GeneArt (Thermofisher Scientific) and cloned into an in-house derivative of the pSC11 vaccina shuttle vector downstream of the synthetic early/late vaccina promoter via Sal1/Not1 restriction sites. The modified pSC11 vector also constitutively expressed green fluorescent protein. Recombinant Modified Vaccinia Ankara (MVA) viruses were generated by transfecting BHK21 cells with the shuttle vector followed by infection with wild-type MVA. Recombinant viruses were then isolated by repeated rounds of plaque purification. Recombinant plaques were identified by visually inspecting infected cell monolayers for GFP positivity. Protein expression was confirmed by western blotting of infected cell lysates using an anti-HA antibody as described [69]. Virus seed stocks and working stocks were prepared by infecting primary chicken embryo fibroblasts at low multiplicity of infection (moi) and titres determined by infecting monolayers with different dilutions of the virus stocks.

## Preparation of synthetic epitope peptides

The original aim of the work was to screen for HLA-class I epitope peptides. Peptide sequences were therefore designed based on analysing the B95.8 strain sequence of each protein using three epitope peptide prediction algorithms (http://www.syfpeithi.de/Scripts/MHCServer.dll/EpitopePrediction.htm; http://www-bimas.cit.nih.gov/molbio/hla_bind/; http://tools.immuneepitope.org/main/). The highest ranked peptides predicted by each algorithm to bind HLA-A2 or HLA-B7 were prioritised for analysis. Starting from the predicted 9-mer sequence, additional flanking residues were added such that the 9-mer formed the core of a 20-mer peptide. Peptides were then synthesised by Alta Bioscience (University of Birmingham, U.K.) and dissolved in DMSO to prepare stock solutions at a final concentration of 5mg/ml. For tetramer production high purity (90–95% purity) peptides were purchased from Peptide 2.0 and reconstituted in DMSO at 20mg/ml.

## Screening for T-cell responses using recombinant MVA viruses or synthetic peptides in IFNγ ELISpot assays

Peripheral Blood Mononuclear Cells (PBMCs) were tested in IFNγ ELISpot assays (Mabtech) as previously described [70]. Two approaches were used. First, in *ex vivo* assays PBMCs were infected with recombinant MVA viruses at an moi of 3 as described [69], washed and then added to wells of an ELISpot assay plate that had been pre-coated with IFNγ capture antibodies and blocked. Typically, cells were tested in triplicate wells with $3x10^5$ cells added to each well. Second, cultured ELISpot assays were performed using synthetic peptides. PBMCs were exposed to small pools of peptides (2–3 peptides/pool) at a final concentration of 2µg/ml for each peptide in separate wells of a 96-well round bottom plate. Following 7-days of *in vitro* culture, cells were washed thrice with media and transferred to replicate wells of an ELISpot plate prepared as above. The cells were then exposed to: i) a fresh aliquot of the original stimulating peptide, ii) an equivalent volume of DMSO solvent as a negative control, or iii) phytohemagglutinin (10µg/ml) as a positive control. For both the *ex vivo* and cultured ELISpot assays, the cells were cultured in the ELISpot plate overnight at 37˚C in 5% $CO_2$ and then processed to detect IFNγ according to the manufacturer's instructions. Spots were visualised using an alkaline phosphatase substrate kit (Bio-Rad) and counted using an automated plate counter (AID). Final counts were normalised to the DMSO negative control wells. In all experiments, results from ELISpot assays are expressed as spot-forming cells (SFC) per million PBMCs.

## T-cell cloning and characterization

T-cells were cloned by limited dilution cloning as previously described [6]. Briefly, PBMCs were stimulated with antigen and cultured *in vitro* for seven days in T-cell media (LCL media with 10units/ml IL2 and 5% human serum replacing FCS). Antigen specific T-cells were then enriched for limited dilution cloning following positive selection using an IFNγ cell enrichment and detection kit (Miltenyi Biotec). For the generation of CD8+ T-cell clones, PBMCs were first depleted of CD4+ T-cells using anti-CD4 Dynabeads (Invitrogen) prior to stimulation and 20 ng/ml IL-7 and 50 U/ml IL2 were added to the culture media. Growing T-cell microcultures were screened for antigen specificity by adding the original stimulus to a sample of each culture and measuring IFNγ production by ELISA.

The minimal epitope recognized by T-cells was defined experimentally by testing them on HLA-matched LCL cells pre-exposed to a panel of shorter peptides of interest that spanned the original stimulating 20mer peptide and the flanking 10 amino acids. For CD4+ T-cell clones we used peptides overlapping by 5 aa. The closed pocket of the HLA-I molecule allows more precise delineation of CD8+ T-cell epitopes and here we used peptides overlapping by 1 aa and varying in length from 8–13 amino acid. Functional avidity was measured in peptide titration assays using each T-cell clone's optimal epitope peptide and was defined as the peptide concentration eliciting 50% of maximal IFNγ [24]. HLA restriction was determined by testing each T-cell clone against the autologous and a panel of partly HLA-matched LCLs pulsed with epitope peptide. T-cell clones were tested for recognition of HLA-matched LCLs transformed with the B95.8 or a recombinant EBV strain that lacked the BZLF1 gene (termed BZLF1-KO LCLs). The T-cell receptor V-beta region of selected T-cell clones was amplified by nested RT-PCR as described [71] except 35 cycles of amplification were used for each round of amplification. Following the second round of PCR, amplicons were purified by agarose gel electrophoresis and DNA sequenced using the second round TCR constant region primer and an M13F(-47) primer (Source Bioscience).

## Assessment of functional T-cell responses to EBV capsid proteins by flow cytometry

B-cells were purified from PBMCs, infected with recombinant MVA virus expressing invariant chain targeted ovalbumin, BcLF1 or BORF1 and then incubated overnight to allow antigen to be expressed. The next day the infected B-cells were washed and mixed with the B-cell depleted PBMCs at a ratio of 1:10 (B-cell:PBMC) in LCL media in a V bottom 96 well plate (typically 1-2x10^6 cells/well). Monensin (0.25ug/ml) and anti-CD107a-FITC (Beckton Dickinson) were added for 1hr then brefeldin A (5ug/ml) was added. After 4-5hrs the cells were stained for 30min with CD3-, CD4- and CD8-specific antibodies along with a fixable live dead stain. The cells were washed, fixed and permeabilized using a Fix&Perm kit (Life Technologies) and stained with antibodies specific for IFNγ and, in experiments assessing polyfunctionality, antibodies specific for IL2, TNFα and GM-CSF. Cells were washed and analysed using an LSR-II flow cytometer (BD Biosciences). All data were processed using FlowJo software (TreeStar).

## $^{51}$Cr cytotoxicity assay

Cytotoxicity was measured using chromium release assays. HLA matched B95.8 or BZLF1-KO LCL cells were labelled with $^{51}$Cr (Amersham, Germany) for 1 hour. When required, epitope peptides were loaded onto LCLs during the labelling step. Cells were washed thrice and resuspended in RPMI supplemented with 100 IU/ml penicillin, 100 mg/ml streptomycin, and 10% FCS and re-suspended. Effector cells were plated to give required effector:target ratios. Targets

were also plated out with the addition of media only (spontaneous release) or 1% SDS (maximal release). Following 5hr incubation, supernatant was removed and analysed on a Packard Cobra II D5010 Scintillation Gamma Counter. Results are expressed as percentage specific lysis calculated using the following formula: (experimental release-spontaneous release) / (maximal release-spontaneous release) X 100%.

## B-cell infection and time course assays

Wild type 2089 EBV was purified using sucrose gradients and quantified as described [72]. Infection of primary B-cells by purified virus was assessed 72hrs after infection by intracellular staining for EBNA-2 (antibody clone PE-2) followed by a PE-labelled secondary antibody and analysis on an Accuri flow cytometer (Becton Dickinson). Purified B-cells, either unmanipulated, infected with virus at 12 virus particles (vp)/cell or pulsed with epitope peptide were added to wells of a 96 well round bottom plate ($5x10^4$ cells/well). T-cells from a clone were added to appropriate wells ($5x10^3$ T-cells/well) and the wells topped up to a final volume of 200μl with media such that they contained 20U/ml IL2. Every 24hrs over the first 5 days, 100ul of culture media was collected and stored and fresh media added. Culture media was also collected on days 7, 10 and 14. At the end of the experiment the quantity of IFNγ in the culture media samples was measured by ELISA. In a second series of experiments, B and T-cells were prepared as above but were placed into separate wells on the plate. At each 24hr time point, a well of T-cells and B-cells were washed and combined together into a single well. Supernatant was then collected from each combined well 24hrs later and IFNγ measured by ELISA.

## B-cell outgrowth assays

B-cells, infected with purified 2089 EBV as above, were added to wells a 96 well round bottom plate ($1x10^4$ cells per well). Peptide pulsed B-cells or B-cells alone were also added to wells in each experiment as controls. Varying numbers of T-cells from a T-cell clone were then added to triplicate wells to establish B-cell:T-cell ratios from 1:1 to 128:1. In some experiments 100vp/cell was used and CSA added to a final concentration of 0.25 μg/ml in some wells of the plate. Outgrowth was scored visually following 21–28 days of culture at 37˚C 5%$CO_2$.

## MHC-II tetramer staining assays

MHC class-II tetramers were purchased from the Tetramer Core Laboratory, Benaroya Research Institute, Seattle, WA. Tetramer staining was performed as previously described [5]. Briefly, cryopreserved PBMCs were thawed, washed in warmed LCL media and re-suspended in 50μl of human serum. A pre-titrated volume of tetramer was then added for 2hrs at 37˚C. The PBMCs were then washed and stained with either LIVE/DEAD stain (Life Technologies) or Green Zombie dye (Biolegend) depending on the experiment followed by appropriate surface antibodies. In some experiments PBMCs were stained intracellularly for perforin and granzyme B as described [44].

## Statistical analysis

Statistical tests, including normality tests, were performed as indicated using GraphPad Prism v9 software. Only results found to be significant ($p<0.05$) are displayed.

## Supporting information

**S1 Fig. Identification and characterization of capsid antigen specific T-cell clones.** Schematic representation of the process of T-cell clone characterisation. A) Donor response to

peptide pools was identified by *ex vivo* IFNγ ELISpot, image shows response from $0.3 \times 10^6$ PBMC per well, to peptide pool or DMSO negative control. T-cell clones were then produced by limiting dilution cloning. B) Expanded T-cell clones were tested against the three individual peptides contained within the original stimulating pool (pool 12) to identify the peptide mediating the response and to ensure specificity. C) Clones were then tested against overlapping peptides to define the minimal and optimal peptide. D) Whether clones were CD4+ or CD8+ T-cells was determined by flow cytometry. E) HLA-restriction was determined by stimulation of T-cell clones with peptide pulsed autologous or partially HLA-matched LCLs. The example shows the response elicited by the autologous donor LCL (top, HLA-class II type: DR1, DR15, DR51, DQ5, DQ6) and five partially matched LCLs. For the latter only the HLA-alleles shared with the autologous LCL are shown. F) To determine functional avidity, the concentration of peptide eliciting 50% of the maximal T-cell activity, T-cell clones were exposed to increasing concentrations of the optimal peptide. For panels B, C and E bars indicate mean±SD.
(TIF)

**S2 Fig. Expression of perforin and granzyme B by capsid antigen specific T-cell clones.** Selected examples of perforin and granzyme B expression in CD8+ (SMF and RAL) and CD4+ (NIL and SMY) capsid specific clones, as measured by flow cytometry. Median Fluorescence Intensity (MFI) values are inset.
(TIFF)

**S3 Fig. NIL tetramer gating and testing.** A) Example of the gating strategy used for identification of tetramer positive CD4+ T-cells. First, doublets were excluded, followed by selection of T-cells (CD3+) with exclusion of CD14+, CD19+, viability dye+ cells (Dump). Lymphocytes, based on SSC vs FSC, and subsequently CD3+CD4+ T-cells were then gated before final tetramer-positive cell identification. B) Tetramer was tested using irrelevant T-cell line unspiked or spiked with T-cells from a NIL-specific T-cell clone (spiked at a 1:4 ratio of NIL clone to irrelevant T-cells) to test tetramer specificity, shown are T-cell clones stained with the BORF1 NIL tetramer. C) Tetramer was then tested on PBMC from a donor with a known NIL-specific response alongside an irrelevant tetramer, which was used to set gates. Shown is an example of BORF1 NIL tetramer staining. D) Linear correlation of the percentage of BORF NIL tetramer stained CD4+ T-cells and the frequency of CD4+ T-cells responding to stimulation with NIL peptide in the same individuals using PBMC from 4 donors.
(TIF)

**S4 Fig. The dynamics of the CD4+ T-cell response to capsid antigens from acute infection to convalescence in two IM patients.** Shown are the frequency of NIL (red) and other tetramer (as indicated) positive CD4+ T-cells during IM (Week 0) and into convalescence (second and third timepoint). Time is relative to the first sample.
(TIF)

**S1 Table. T-cell receptor V-beta sequences of NIL-specific CD4+ T-cell clones.** Analysis of VDJ gene usage was performed using the international immunogenetics information system V-Quest tool (http://www.imgt.org).
(DOCX)

## Acknowledgments

We thank Dr Clare Shannon-Lowe (University of Birmingham) for her assistance with purifying and quantitating EBV preparations.

## Author Contributions

**Conceptualization:** James E. Turner, Graham S. Taylor.

**Data curation:** Tracey A. Haigh.

**Formal analysis:** Alexander C. Dowell, Tracey A. Haigh, Graham S. Taylor.

**Funding acquisition:** Graham S. Taylor.

**Investigation:** Alexander C. Dowell, Tracey A. Haigh, Gordon B. Ryan, James E. Turner.

**Methodology:** Alexander C. Dowell, James E. Turner, Graham S. Taylor.

**Project administration:** Graham S. Taylor.

**Resources:** Gordon B. Ryan, James E. Turner, Heather M. Long.

**Supervision:** Heather M. Long, Graham S. Taylor.

**Visualization:** Alexander C. Dowell, Graham S. Taylor.

**Writing – original draft:** Alexander C. Dowell, Graham S. Taylor.

**Writing – review & editing:** Tracey A. Haigh, James E. Turner, Heather M. Long.

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
