## [Decision Letter · Decision Letter 0]

25 Jun 2021

Dear Dr Dowell,

Thank you very much for submitting your manuscript " Cytotoxic CD4 T cells specific for EBV capsid antigen BORF1 are maintained in long-term latently infected healthy donors. " for consideration at PLOS Pathogens. As with all papers reviewed by the journal, your manuscript was reviewed by members of the editorial board and by several independent reviewers. In light of the reviews (below this email), we would like to invite the resubmission of a significantly-revised version that takes into account the reviewers' comments.

We cannot make any decision about publication until we have seen the revised manuscript and your response to the reviewers' comments. Your revised manuscript is also likely to be sent to reviewers for further evaluation.

Sincerely,

Christian Munz

Associate Editor

PLOS Pathogens

Erik Flemington

Section Editor

PLOS Pathogens

Kasturi Haldar

Editor-in-Chief

PLOS Pathogens

orcid.org/0000-0001-5065-158X

Michael Malim

Editor-in-Chief

PLOS Pathogens

orcid.org/0000-0002-7699-2064

Reviewer's Responses to Questions

**Part I - Summary**

Reviewer #1: In this manuscript from Dowell and colleagues the investigators report on T cell immunity to a subset of EBV late structural antigens, which are present in the virion and hence theoretically available as an antigen source early during infection. In an elegant series of experiments performed using MVA vectors expressing individual antigens as a source of stimulus, and subsequently in-depth studies of the phenotypic and functional characteristics of T cell clones, the authors characterize a series of different reactivities. Furthermore, they assess the biologic relevance of such responses by examining the potential of reactive cells to control the outgrowth of EBV transformed cells. Finally, they describe in detail an interesting cytotoxic CD4 response mapping to BORF1. This group has deep expertise in EBV, as well as characterizing and analyzing T cell immunity both in the setting of healthy individuals during their primary exposure (infectious mono) and evolution to memory, as well as in those with EBV-associated malignancies.

Reviewer #2: In this paper, Dowell et. al. describe some findings of their studies into cellular immunity against some structural antigens of the Epstein-Barr virus. 10 EBV capsid protein-derived epitopes are identified and T cells studied for function and phenotype. The findings of the work are documented well, albeit some figures could be worked on to improve lucidity and avoid repetition of presentation. Even though MHCII molecules restricting the epitopes are described using non-current nomenclature, the information on previously undescribed and immunologically relevant epitopes is worthy of documentation and therefore worth considering for publication.

Reviewer #3: The authors describe the discovery of novel CD4+ T cell epitopes that target BORF1 and other proteins of EBV. Most interesting is the cytotoxic capacity of the BORF1 specific T cells against EBV-transformed B cells. While EBV has bene studied exhaustively for the present of T cell epitopes, there is some novelty in the discovery of CD4+ T cell epitopes from BORF1, particularly if they have the capacity to recognize and kill cells early after entering the lytic phase.

The major limitation is that it is not really clear whether or not cytotoxic activity is a common feature of T cells specific for the BORF1 epitope NIL. The use of T clones provides a very clean approach to study cytotoxic capacity but it provides no insight the diversity of the T cell response to the NIL peptide, nor does it demonstrate if cytotoxic capacity is a common feature of all clones specific for NIL.

**Part II – Major Issues: Key Experiments Required for Acceptance**

Reviewer #1: 1. Please add some context in the discussion helping the reader to evaluate the relative biological importance of T cell activity to the antigens explored in the current study vs that directed against other latent and lytic antigens (much of which has also been characterized by the same group). It appears that many EBV proteins are immunogenic and induce T cells that are protective in nature – and as such there seems to be a lot of redundancy from the perspective of keeping this virus quiescent. But is there some way to rank these antigens along with other well characterized antigens to determine what is minimally required to control this virus. Given this groups deep expertise it would be helpful if they could begin to frame the discussion.

2. I thought it interesting that the NAI clones were not only cytolytic but also produced the highest concentrations of IFNg (~12ng/ml – day 1 -- comparable to that seen with the gp350 clone). Furthermore, both of these clone specificities were able to arrest the outgrowth of transformed B cells. This contrasted with QPR and SMY (both ~4.5 ng/ml), which recognized LCLs but were not cytolytic or able to prevent B cell outgrowth. Although the authors demonstrate that cytokine was not linked to the cytolytic function of NAI T cells I wonder if they can comment about whether there may be some threshold of activity (IFNg concentration) that may predict effector function.

Reviewer #2: Main issues:

1. The build-up of the story and the discussion of the results fails to do due justice to previous works in the field. This is a major scientific issue and is elaborated to some extent in some of the specific comments below.

2. The authors frequently use terms in the statistical sense but without showing any statistic to that end. e. g. use of the term “significant” in line 228, line 287, line 289, line 369.

3. The discussion fails to actually address most of the findings and therefore, largely reads more or less like a summary of the results. Discussion on the biological significance of the findings of the work would better help to put the findings in context.

4. The paper has some incomplete sentences here and there, as well as grammatical errors. Did the authors send out a preliminary draft version by mistake?

Reviewer #3: The NIL clone shows good degranulation and killing capacity, however I think stronger support for the presence of cytotoxic T cells should be provided through analysis of granzyme and perforin expression in the clones and some comparison with conventional CD8+ CTL provided. Similarly comparative analysis with CD8+ CTL should be provided for context in Figure 5D when staining during primary IM.

How many NIL clones have cytotoxic capacity? Particularly clones that have different TCR. The authors should provide analysis of multiple clones to demonstrate that cytotoxic function isn't merely a feature of one clonotype.

**Part III – Minor Issues: Editorial and Data Presentation Modifications**

Reviewer #1: n/a

Reviewer #2: Additional comments:

1. As a background to the work, the authors state in the abstract “Continued T cell immune responses are necessary to the control of latent infection of which CD4 T cell responses are critical. Previous investigations have predominantly focused on immediate early and early viral proteins, as such relatively little is known regarding the response to late structural antigens.” and “…little is known about the cellular response to….structural antigens”. The validity of these declarations is debatable. CD4+ T cell epitopes from structural antigens of EBV have been described in several publications with the first ones dating over thirty years back e.g. PMID: 2904885, 1988.

2. For the cytotoxic potential of CD4+ T cells against EBV, the authors refer to three publications from 2017, 2019 and 2020 (see line 49), but fail to acknowledge earlier studies e.g. PMID: PMID: 12133989, 2002, PMID: 14581546, 2003. There are further examples that perhaps deserve due mention.

3. “In contrast to other lytic-cycle antigens, structural, capsid and envelope proteins, will by nature, be present at the earliest point of cellular infection. As such we hypothesize that these proteins will provide a potential antigen source for T cell recognition of newly infected cells at an optimal time in the infection cycle.” This hypothesis has in fact already been raised before (see Figure 1, PMID: 16549599, 2006)

4. The antigen BNRF1 has been systematically studied and several epitopes have been reported recently. So line 62 that reads “As the T cell response to many of the structural antigens are poorly defined, we set out to characterize the response to major capsid antigen - BcLF1, Triplex capsid protein 1 - BORF1, triplex capsid protein 2 - BDLF1, and major tegument protein - BNRF1.” may need revision.

5. Line 374: “Typically, the EBV CD4 T cell response, which is generally not cytotoxic…” There are plenty of cytotoxic CD4+ T cells reported, with some reports being around twenty years old.

6. Line 352: “While cytotoxic CD4 responses in vivo have been demonstrated in the context of active infection, they have not been attributed to CD4 cells in latent infection”. Most cytotoxic CD4+ T cells and their target epitopes have been reported from healthy EBV carriers and therefore are by definition described in the context of latent infection.

7. The penultimate sentence in the concluding paragraph line 424: “Taken together the present study has shown that CD4 T cells are functional and suited to early recognition of newly infected B cells”. That CD4+ T cells are suited to early recognition of newly infected (24hr) B cells, is known (PMID: 16549599, PMID: 16549597).

8. Line 400: extending a peptide from a 9-mer to a 10-mer should not be called minimizing.

9. Line 401: The authors acknowledge that they find something in this work that is different from a finding that they have reported before. This honesty is appreciable. But the finding is something that should perhaps be addressed further because inconsistency of findings may discredit the study, which would be a loss for the sake of the other findings.

10. Statistical evaluation would perhaps be of use in improving the validity of generalized conclusions derived from studies in limited numbers of individuals or T cell clones.

11. The ex vivo direct demonstration with tetramers is nice to see. Given that MHCII tetramer staining is a recently developed technology, it would add value to the findings if the authors would show primary data from assay establishment, especially the criteria they used to define background and signal. Same holds for ELISpot assays.

12. Line 337: The authors show data from 7 donors, it appears that 4 of them have CD4+ responses and 5 of them have CD8+ responses against BORF1. It is therefore unclear how the authors come to the statement “These data show that the BORF1 T Cell response is dominated by cytotoxic CD4 T cells.”

13. The finding that different T cell clones recognizing the same antigen respond differently to newly infected cells requires exploration to rule out errors in experiment set up and sampling.

14. Line 243 – The authors make a claim and state that “as a rule”. Whereas the statement may be valid, are the cases studied enough to identify a rule? (Three BcLF1 clones tested, one does not support the stated “rule”.)

15. Line 227: Because exact determination of infectious particles in a virus-containing preparation is challenging and because inter-experimental and inter-laboratory differences are known to exist, it is suggested that the authors avoid stating generally that an moi of 12 is at the cusp of in vitro transformation. Information on how the authors measured viral particle numbers should be included.

16. Line 94: What the authors mean by “selected on GFP” is not clear.

17. Line 309- “varied widely”- would showing coefficient of variance be possible?

18. The use without definition of non-standard terms like “high” in line 299, “substantially different” in line 364 are perhaps better avoided.

19. Line 104 “whole MVA antigen” might be confused to understand antigens of the MVA virus and not the EBV antigens that the authors mean.

20. There are several inconsistencies in writing style and the usage of terms: for example “9-mer” but “20mer”, “CD4+ and CD4”, “maximal release” and “maximum release”, “BZLF1 KO” and “BZ KO”, “MOI” and “moi”, “T Cell” and “T cell”, “BDLF1” and “BDLF-1”, just to name some. The use of a dash between two words in two-word adjectives is also inconsistent. The term “complete media” and “standard new infection assay” are used without defining them.

21. At some places in the “Results” and some places in “Discussion” there are inconsistencies in the usage of past versus present tense form for verbs used for reporting.

22. Does vp/cell mean virus particles per cell? It would help to have it spelled out somewhere.

23. Line 93- inserted “by the restriction site”. Please consider revising.

24. Line 159: “96 round bottom plate”. “well” missing?

25. Line 270- mislabel: Figure 4C not 4B

26. Figure 1A, 5A, 5B- what do the red dashes in the figures represent?

27. Figure 2B: T cell clones produce up to 12.5 pg/ml IFNg? Do you mean ng/ml?

28. Figure 2E- Absolutely no lysis of BZ KO cells? No error bars even? If this is not a mistake, this deserves to be commented on.

29. Fig 3D- The BNRF1 TDA cells produce 2ng/ml of IFN gamma. It is unclear why the authors state in Results that the TDA clone does not recognize infected cells.

30. Figure 4A- Perhaps it would be easier to understand and remember if the ratios were of the type 1:67 rather than 0.015:1

31. Figure 4B- please explain why multiplicity of infection of 10 was chosen, even though you declare in the text that 12 is on the cusp of in vitro transformation.

32. In Figure 4, HLA-matched allogeneic B cells were used as targets. Can the authors exclude that inhibition of outgrowth by BORF1-NIL was not due to alloreactivity?

Information on how HLA-restriction was determined would be appreciated.

33. Figure 7D is hard to follow in grey scale. There are indeed coloured figures in the paper. Any particular reason why 7D had to be in grey scale?

34. Please run a spelling and grammar scan through the document, there are instances not mentioned above where words appear to be missing. For example sentence starting at line 135. Some examples of incomplete sentences: line 37, line 220-221, line 298, line 300.

Reviewer #3: The authors could try and normalize the axes in Fig 3D. It is annoying to compare values of stimulation when the axes are so variable. It would also be nice to see the proportion of T cells that are activated in Figure 3, not just cytokine release. I am surprised more standard ICS assessment isn't used in these assays.

I am not sure of the relevance of Figure 5F. Yes PRS declines but from a much higher starting percentage that possibly includes a higher percentage of effector T cells. This data would be better shown as actual percentages and with more donors that allows for variance to be included. It seems to include only a single donor for each.

Why no PRS data in Figure 6, considering it is used as a comparison in Fig 5?

PLOS authors have the option to publish the peer review history of their article (what does this mean?). If published, this will include your full peer review and any attached files.

Reviewer #1: No

Reviewer #2: No

Reviewer #3: **Yes: **Corey Smith
---

## [Editor Report · Decision Letter 1]

4 Nov 2021

Dear Dr Dowell,

Thank you very much for submitting your manuscript "Cytotoxic CD4 T cells specific for EBV capsid antigen BORF1 are maintained in long-term latently infected healthy donors." for consideration at PLOS Pathogens. As with all papers reviewed by the journal, your manuscript was reviewed by members of the editorial board and by several independent reviewers. The reviewers appreciated the attention to an important topic. Based on the reviews, we are likely to accept this manuscript for publication, providing that you modify the manuscript to include statistical analyses for significant changes and include a statistics section to the methods section.

Sincerely,

Christian Munz

Associate Editor

PLOS Pathogens

Erik Flemington

Section Editor

PLOS Pathogens

Kasturi Haldar

Editor-in-Chief

PLOS Pathogens

orcid.org/0000-0001-5065-158X

Michael Malim

Editor-in-Chief

PLOS Pathogens

orcid.org/0000-0002-7699-2064

Reviewer Comments (if any, and for reference):

Figure Files:

Data Requirements:

Reproducibility:

References:

---

## [Editor Report · Decision Letter 2]

22 Nov 2021

Dear Dr Dowell,

We are pleased to inform you that your manuscript 'Cytotoxic CD4 T cells specific for EBV capsid antigen BORF1 are maintained in long-term latently infected healthy donors.' has been provisionally accepted for publication in PLOS Pathogens.

Best regards,

Christian Munz

Associate Editor

PLOS Pathogens

Erik Flemington

Section Editor

PLOS Pathogens

Kasturi Haldar

Editor-in-Chief

PLOS Pathogens

orcid.org/0000-0001-5065-158X

Michael Malim

Editor-in-Chief

PLOS Pathogens

orcid.org/0000-0002-7699-2064
---

## [Editor Report · Acceptance letter]

3 Dec 2021

Dear Dr Dowell,

We are delighted to inform you that your manuscript, " Cytotoxic CD4+ T-cells specific for EBV capsid antigen BORF1 are maintained in long-term latently infected healthy donors. ," has been formally accepted for publication in PLOS Pathogens.

Best regards,

Kasturi Haldar

Editor-in-Chief

PLOS Pathogens

orcid.org/0000-0001-5065-158X

Michael Malim

Editor-in-Chief

PLOS Pathogens

orcid.org/0000-0002-7699-2064